# Response to self-care practice messages among patients with diabetes mellitus visiting Jimma University medical center facility based cross sectional design application of extended parallel process model

**Mohammed Jemal Abawari** * , **Demuma Amdisa, Zewdie Birhanu, Yohannes Kebede**

Department of Health, Behavior and Society, Faculty of Public Health, Jimma University, Jimma, Ethiopia

* Mahirmohammed159@gmail.com

## Abstract

### Objective

To determine response to self-care practice message among patients with diabetes in Jimma University Medical center based on the Extended Parallel Process Model.

### Design

A facility-based cross-sectional study was conducted.

### Setting

Jimma University Medical Center is found in Jimma town.

### Participants

A total of 343 patients with diabetes participated in the study; making a response rate of 93.9%. All patients with diabetes who were 18 years and above and who were on follow up and registered were included in the study and those with Gestational DM were excluded.

### Data analysis

Multivariable logistic regression analysis.

### Result

Responsive respondents scored high in self-care practice score as compared to other respondents. educational status, information sources, knowledge, and preferred message appeals were independent predictors of controlling the danger of diabetes.

**Data Availability Statement:** All relevant data are within the manuscript and its Supporting Information files.

**Funding:** The author(s) received no specific funding for this work.

**Competing interests:** The authors have declared that no competing interests exist.

## Conclusion

There is a significant gap in controlling the danger of diabetes. Variables like the level of education, knowledge of diabetes mellitus, information sources, and message appeals were independent predictors of controlling the danger of diabetes. Designing message having higher efficacy while maintaining the level of threat is the best that fits the existing audience's message processing to bring about desired diabetic self-care Practice.

## Background

Globally, diabetes is one of the top 10 causes of death. The most recent IDF atlas 2019 points worldwide there are 351.7 million people of working age (20–64 years) with diagnosed or undiagnosed diabetes in 2019. In Africa alone, 19.4 million people are living with diabetes [1, 2]. In Ethiopia, the magnitude of diabetes is increasing; according to the WHO report, the number of cases was 800 000 in 2000 and is rising to an estimated 1.8 million by 2030 [3–5].

Behaviors undertaken by people with or at risk of diabetes to successfully manage the disease on their own is known as self-care practices which include four main domains: sustaining appropriate dietary practice, engaging in regular physical exercise and self-monitoring of blood glucose levels and foot care [6, 7]. Self-care behavior is associated with good glycemic control with a mean of HbA1c level changed from 8.3% to 7.3% reduction of complications and improvement in the quality of life [8, 9].

Despite the benefits of self-care practice in reducing diabetes complications and improvement of lifestyle, studies done in both developed and developing countries showed poor self-care practice. One reason for this may be problems of communication which is important in influencing perception, attitude intention, and behavior change [10–15]. The EPPM is one of persuasive communication model which helps to see the effect of message processing in developing realistic risk perceptions and actionable information about how to reduce risk [16].

Even though studies are conducted in identifying communication efforts and persuasiveness of the message conveyed to the targeted individuals in different parts of the world little is known on how patients' access, process, and responds to diabetic self-care messages in Ethiopia. Therefore, this study fills these gaps by assessing response to self-care message among Diabetic Patient in Jimma University Medical Center Based on EPPM.

### Specific objectives

- To assess self-care message response among patient with diabetes
- To analyze predictors of response to diabetes self-care message
- To assess the association between response to message and diabetes self-care practice

## Methods

### Study design

- The facility-based cross-sectional study design was carried out from April 12 to May 25, 2020.

## Setting

The study was conducted in JUMC from April 12 to May 25, 2020. JUMC is found in Jimma town which is located in the Oromia region, Southwest Ethiopia, at 343 Km from Addis Ababa, the capital city of Ethiopia. The total numbers of registered diabetes patients on follow-up are 3578 in JUMC.

**Participants.** *Source population.*

- All patients with diabetes that are 18 years and above and attending follow up at diabetes mellitus clinics in JUMC.

*Study population.*

- Selected patients with diabetes who were 18 years and above who visited JUMC during the study period.

## Eligibility criteria

**Inclusion criteria.**

- All patients with diabetes who were 18 years and above and who were on follow up and registered

**Exclusion criteria.**

- Gestational DM

## Variables

**Perceived threat.**   Cognitions about danger or harm that exists in an environment. Perceived threat comprises two underlying dimensions: perceived severity and perceived susceptibility [17, 18].

**Perceived severity.**   Beliefs about the significance or magnitude of the diabetes complication [17, 18].

**Perceived susceptibility.**   Beliefs about one's risk of experiencing diabetes complication [17, 18].

**Perceived efficacy.**   It is Cognitions about effectiveness, feasibility, and ease with which the recommended response impedes or averts a threat. It contains two underlying dimensions: response efficacy and self-efficacy [17, 18].

**Perceived self-efficacy.**   Beliefs about one's ability to perform the diabetes self-care to avert the diabetes complication [17, 18].

**Perceived response efficacy.**   Beliefs about the effectiveness of the diabetes self-care in deterring or avoiding the diabetes complication [17, 18].

**Source of information.**   Asks the respondents to choose between different sources of information about diabetes self-care practice.

Fear appeal:—are message designed to arouse fear in people

Dramatic appeal:—are message designed to entertain people.

**Cues to action.**   Are strategies to activate readiness includes events, people, or things that move people to change their behavior [19].

**Danger control responses.** It is a self-protective motivation. It includes Belief, attitude, intention, and behavior changes(diabetes self-care) under a message's recommendations [17, 18].

When the critical value is positive the individual is in danger control response [20].

**Fear control responses.** It is a defensive motivation. Coping responses that diminish fear such as defensive avoidance, denial, and reactance (including issue and message derogation and perceived manipulative intent) [17, 18]. When the critical value is $\leq 0$ an individual is in fear control response [20].

**Quadrant I: (Responsive respondents).** People taking protective action against health threat (diabetes complication) [21].

**Quadrant II: (Fear control respondents).** People in denial about health threat (diabetes complication), reacting against it [21].

**Quadrant III: (Proactive respondents).** Lesser Amount of Danger Control:-People taking some protective action, but not really motivated to do much [21].

**Quadrant IV: (No response respondents).** People not considering the threat (diabetes complication) to be real or relevant to them; often not even aware of threat [21]

**Diabetes self-care.** The four sub-scale domains include diet, physical activity, blood glucose testing, and foot-care.

## Measurements

**Perceived threat.** It was measured with 8 items adopted from the RBD which is validated tool when applying EPPM in different contexts. The score of weighted perceived susceptibility and perceived severity was summed up to form the score of weighted perceived threat. The response was summed up and standardized with a response ranging from 0–100 and the score was treated as a continuous variable and higher score shows better perceived threat.

**Perceived efficacy.** It was measured with 8 items adopted from the RBD which is validated tool when applying EPPM in different contexts. A weighted score of perceived self-efficacy and perceived response efficacy was summed up to form a score of weighted perceived efficacy. The response was summed up and standardized with a response ranging from 0–100 and the score was treated as a continuous variable and higher score shows better perceived efficacy.

**Knowledge about diabetes.** It was measured using seventeen items with a yes and no response which was summed up and weighted with a response ranging from 0–100 and the score was treated as a continuous variable.

**Source of information.** The score was summed up to form composite score and was treated as continuous variable.

**Preferred message appeals.** Respondents were asked to choose between two types of message appeals i.e., Fear appeal (message designed to arouse fear in people) and dramatic appeal (message designed to entertain people).

**Critical value (discriminating value).** Obtained by subtracting weighted perceived threat score from weighted perceived efficacy score. Respondents who scored above 1 are danger control respondents [22].

**Diabetes self-care.** Validated Summary of Diabetes Self-Care Activities (SDSCA) questionnaire was used to measure diabetic self-care practice. The questionnaire comprises of 10 items with four sub-scale domains. The four sub-scale domains include diet, physical activity, blood glucose testing, and foot-care. The SDSCA measures the frequency of performing diabetes self-care activities in the last 7 days. Response choices range from 0 to 7. The mean score of diabetic self-care was calculated and those who scored above the mean were categorized as having good diabetes self-care practice [23].

**Quadrant I: (Responsive respondents).**    These are respondents who scored above the median for both perceived efficacy and threat i.e. these are people having high efficacy and high threat.

**Quadrant II: (Fear control respondents).**    These are respondents who scored below the median for perceived efficacy and above the median, for perceived threat i.e. these are people having low efficacy and high threat.

**Quadrant III:(Proactive respondents.**    These are respondents who scored above the median for perceived efficacy and below-median for perceived threat i.e. these are people having high efficacy and low threat.

**Quadrant IV: (No response respondents).**    These are respondents who scored below the median for both perceived efficacy and threat i.e. these are people having low efficacy and low threat.

*Bias*. Since the data collection method was self-report rather than direct observation of the patient's self-care practice this may result in courtesy bias. However, efforts were made to minimize the bias by recruiting data collectors from other department and telling the participants about the anonymity of the data.

*Study size*. The sample size was determined using a single population proportion formula. Accordingly, the formula for sample size determination is: n = (Z$\alpha$/2)2 [(p1q1)/ (d) 2], where n denotes the sample size, Z $\alpha$/2: standard normal score at a 95% confidence interval = 1.96, P: the proportion of danger control response (50%, no previous study found), and D: marginal error of 5% was used. Hence, after adjusting for the total registered patients in the hospital which is 3578 and 5% non-response rate the calculation yielded a sample size of 365 visitors. Every two patients were selected using a systematic random sampling technique until the required sample size was fulfilled by considering the flow of patients in forty-five days.

*Data processing and analysis*. Data analysis was managed using SPSS version 23.0. Before further analysis normality curve and tests of homogeneity of variances were checked, Multicollinearity was checked using VIF and Model fitness was checked by Hosmer and Lemeshow goodness of fit test. Independent sample t-test and analysis of variance (ANOVA) were done to test differences in diabetes self-care practice difference by quadrants. A median split was performed on both threat and efficacy; in both cases respondents at or below the median were placed into the "low" group, and respondents above the median were placed in the "high" group. Predictors of controlling the danger of diabetes were performed using logistic regression. Reliability of each construct was measured using Cronbach's alpha (ranges from 0.79 to 0.88). Variables with p value < 0.25 were selected as a candidate variable for multivariable logistic regression. Finally, only significant variables (P value < 0.05) was retained in the model.

## Result

### Socio-demographic characteristic of diabetic patients

A total of 343 diabetic patients participated in the study; making a response rate of 93.9%. The mean age of the respondents was 48.1 (±14.6) years old. More than half of 182 (53.1%) were male respondents. The major share of participants were followers of Muslim religion, 176 (51.3%); belong to Oromo ethnic group, 224(65.3%); married, 243(70.8%); and attended primary schools or less, 110 (61.2%) (see Table 1).

### Message exposure to diabetes self-care message among diabetic patient

Regarding message exposure, the majority of 330 (96.2%) of the respondents heard about self-care practice in the past six months. Regarding the preferred channels to see or hear about

**Table 1. Socio-demographic characteristic of patients with diabetes in Jimma University medical center, Ethiopia April 12-May 25 2020 (n = 343).**

| Variables | Categories | Frequency &Percentages (%) |
|---|---|---|
| **Age of respondents** | 18–29 | 47 (13.7) |
| | 30–44 | 71 (20.7) |
| | 45–60 | 160 (46.6) |
| | >60 | 65 (19) |
| **Sex** | Male | 182 (53.1) |
| | Female | 161 (46.9) |
| **Marital status** | Married | 243 (70.8) |
| | Single | 58 (16.9) |
| | Divorced | 21 (6.1) |
| | Widowed | 21 (6.1) |
| **Religion** | Muslim | 176 (51.3) |
| | Orthodox | 104 (30.3) |
| | Protestant | 50 (14.6) |
| | Catholic | 13 (3.8) |
| **Ethnicity** | Oromo | 224 (65.3) |
| | Amhara | 31 (9.0) |
| | Kaffa | 26 (7.6) |
| | Gurage | 24 (7.0) |
| | Dawuro | 22 (6.4) |
| | Others | 16 (4.7) |
| **Educational status** | Cannot read and write | 104 (30.3) |
| | Primary school (1–8) | 106 (30.9) |
| | Secondary school (9–12) | 75 (21.9) |
| | College and above | 58 (16.9) |
| **Occupation** | Government employee | 82 (23.9) |
| | Housewife | 75 (21.9) |
| | Merchant | 65 (19.0) |
| | Student | 61 (17.8) |
| | Farmer | 60 (17.5) |
| **Income (ETB)** | <500 | 114 (33.2) |
| | 500–1500 | 64 (18.7) |
| | 1501–3000 | 86 (25.1) |
| | >3000 | 79 (23) |
| **Distance to the nearest health facility** | < 5km | 190 (55.4) |
| | 5km and above | 153 (44.6) |
| **Duration since treatment** | 1–5 | 205 (59.8) |
| | 6–10 | 97 (28.3) |
| | above 10 | 41 (12.0) |
| **Types of diabetes** | Type 1 | 81 (23.6) |
| | Type 2 | 262 (76.4) |

diabetic self-care practice two-third (68.5%) of the respondents prefer television followed by radio (32. 4%). most of 225 (65.6%), the respondents prefer a message that is dramatic/funny.

Regarding specific self-care practice and answering more than one answer was possible, from all the participant majority 318(92.7%) heard about dietary practice, while 235(68.1) heard about foot care, 233(67.9%) and 108(31.4%) heard about regular physical exercise and

**Table 2. Knowledge about diabetes mellitus of patients with diabetes in Jimma University medical center, Ethiopia April 12-May 25 2020 (n = 343).**

| Variables | | Response categories | |
|---|---|---|---|
| | | Yes (%) | No (%) |
| Diabetes is a chronic disease | | 271 (79) | 72(21) |
| Diabetes is not curable | | 270(78.7) | 73(21.3) |
| Ways of controlling diabetes | Diet only | 318(92.7) | 25(7.3) |
| | Regular physical exercise | 233(67.9) | 110(32.1) |
| | Taking drugs | 211(61.5) | 132(38.5) |
| | Measuring blood glucose | 169(49.3) | 174(51.7) |
| Signs of diabetes mellitus | Polyphagia | 288(84) | 55(16) |
| | Polydipsia | 239(69.7) | 104(30.3) |
| | Polyuria | 227(66.2) | 116(33.8) |
| | Weakness | 193(56.3) | 150(43.7) |
| Complications of diabetes mellitus | Foot ulcer/Gangrene | 228(66.5) | 115(33.5) |
| | Kidney problems | 226(65.9) | 117(34.9) |
| | Eye problems | 206(60.1) | 137(39.9) |
| | Heart problems | 188(54.8) | 155(45.2) |
| | Hypoglycemia | 164(47.8) | 179(52.2) |
| | Hypertension | 161(46.9) | 182(53.1) |
| | Nerve problems | 151(44) | 192(56) |

self-blood glucose monitoring respectively. Most of the respondents received information from 2–3 sources.

## Knowledge about diabetes mellitus and cues to action related to DM of respondents

Concerning knowledge on general diabetes mellitus majority of 271(79%) knew diabetes is a chronic disease. Comprehensive knowledge of general diabetes mellitus means score 59(±20.7) and cues to action with a mean score of 1.9(±1.1) (see Table 2).

## Mean, standard deviation and reliability Scores of constructs of EPPM

Regarding perceptions, respondents had a Perceived threat mean score of 79.8(SD ±10.7) and a perceived efficacy mean score of 79.2 (SD±13.7). Cronbach's α score for all the constructs were > 0.7 (see Table 3).

**Table 3. Respondents mean, standard deviation and reliability scores of constructs of the extended parallel process model in Jimma University medical center, Ethiopia April 12-May 25 2020 (n = 343).**

| Variables | Number of items | Response Range | Mean(±SD) | Cronbach's α |
|---|---|---|---|---|
| Perceived threat(overall) | 8 | 0–100 | 79.8(±10.7) | 0.884 |
| Perceived susceptibility | 4 | 0–100 | 77.8(±12.8) | 0.808 |
| Perceived severity | 4 | 0–100 | 81.8(±12.1) | 0.791 |
| Perceived efficacy | 8 | 0–100 | 79.2(±13.7) | 0.884 |
| Perceived response efficacy | 4 | 0–100 | 76.1(±15.5) | 0.876 |
| Perceived self-efficacy | 4 | 0–100 | 82.3(±14.5) | 0.791 |

**Table 4. Response to self-care message among patients with diabetes by efficacy threat interaction in Jimma University medical center, Ethiopia April 12-May 25 2020 (n = 343).**

| PERCEIVED THREAT | PERCEIVED EFFICACY | |
|---|---|---|
| | **High Efficacy (%)** | **Low Efficacy n (%)** |
| **High Threat n (%)** | 73 (21.3%) **Quadrant I**: Responsive (Danger Control) | 61 (17.8) **Quadrant II**: Avoidant (Fear Control) |
| **Low Threat n (%)** | 80 (23.3%) **Quadrant III**: Pro-Active (Small Danger Control) | 129 (37.6) **Quadrant IV**: No Response (indifferent) |
| **Control response based on DV** | 173 (50.4%) (Fear Control Response) | 170 (49.6%) (Danger Control Response) |

## Response to self-care message among patients with diabetes by efficacy threat interaction

Among the respondents 73(21.3%) were responsive respondents, 61(17.8%), were fear control respondents, 80(23.3%) were proactive and (37.6%) were no response respondents. Moreover, 173(50.4%) of the respondents belong to fear control response based on discriminatory value. (see Table 4).

## Relationship between diabetes self-care messages and self-care practices

In this study Control response based on discriminatory value best predicts actual self-care practice(r = 0.487) as compared to control response based on quadrants (r = 0.314) using Pearson correlation coefficients. more over 126(72.8%) of fear control respondents were in poor diabetes self-care practice and 107(62.9%) of danger control respondents were in good in self-care practice. (see Table 5).

## Diabetes self-care practice of diabetic patients

Among all respondents of this study, more than half of them 189 (55%) are in poor diabetic self-care practice (Fig 1).

## Difference in mean diabetic self care practice by efficacy threat interaction

Analysis of variance (ANOVA) showed that mean self-care practice score was significantly different by efficacy/ threat interaction (quadrants); for example, post hoc test using Bonferroni method showed that responsive respondents scored high in mean diabetes self-care practice as compared to fear control and no response respondents additionally proactive respondents scored high in mean diabetes self-care practice as compared to fear control and no response respondents (Table 6).

**Table 5. Relationship between responses to diabetes self-care messages and self-care practices of patients with diabetes in JUMC, Ethiopia April 12-May 25 2020 (n = 343).**

| Response category | Self-care practice category | |
|---|---|---|
| | **Poor self-care (%)** | **Good self-care (%)** |
| Fear Control | 126(72.8) | 47(27.2) |
| Danger Control | 63(37.1) | 107(62.9) |

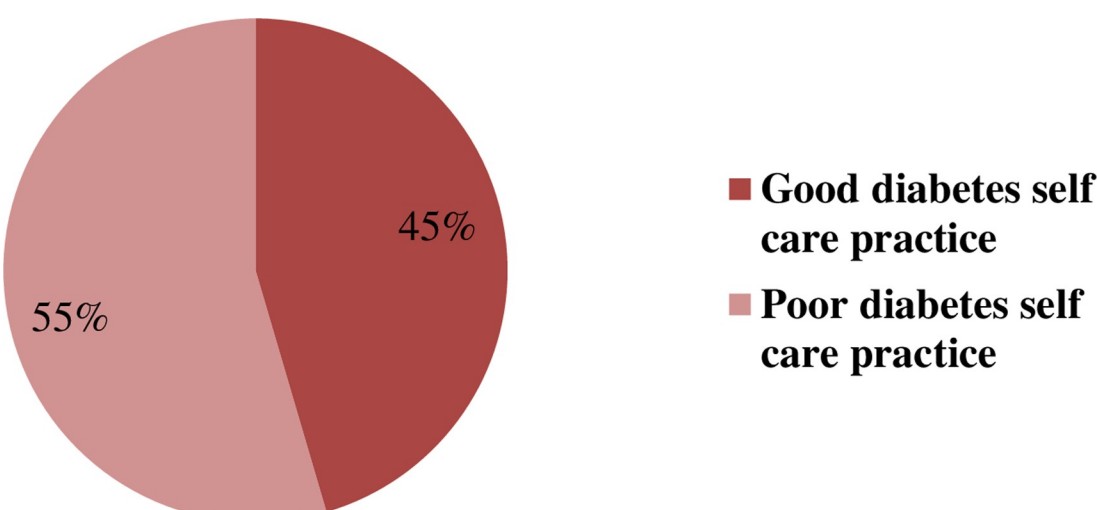

**Fig 1. Showing diabetes self-care practice of patients with diabetes in Jimma University medical center, Ethiopia April 12-May 25 2020 (n = 343).**

### Defensive avoidance scores of diabetes patients

Regarding Defensive avoidance of diabetic complications of respondents (fear control response) participants scored a mean defensive avoidance score of 47.9(SD ±21.4).

### Difference in defensive avoidance (fear control response) by efficacy threat interaction (quadrants)

Analysis of variance (ANOVA) showed that the score of defensive avoidance was significantly different by efficacy/threat interaction; for example, post hoc test using Bonferroni method showed that fear control respondents scored high in mean defensive avoidance score compared to responsive and proactive respondents (Table 7).

### Predictors of response to self-care message among patients with diabetes

The result of the multivariate logistic regression model revealed that educational status, information sources, knowledge of diabetes mellitus, preferred message appeals were predictors **of Response to self-care message among patients with diabetes**

The study revealed respondents who completed college and university were 4.8 times more likely to respond to self-care message in favor of controlling the danger of diabetes compared to those who cannot read and write [AOR = 4.8(2.016, 11.612)] and those who prefer

**Table 6. Showing difference in mean diabetic self care practice by efficacy threat interaction (Quadrants) of patients with diabetes in JUMC, Ethiopia April 12-May 25 2020 (n = 343).**

| ANOVA test statistics | Post hoc Method | Reference groups | Comparison group | Mean difference | P-value | 95%CI |
|---|---|---|---|---|---|---|
| F = 18.261 | Bonferroni | Responsive | Proactive | 0.22 | 0.711 | (-0.23,0.68) |
| df = 3 | | | Fear Control | 0.79 | <0.001 | (0.29,1.29) |
| P-value <0.001 | | | No response | 1.03 | <0.001 | (0.62,1.45) |
| | | Proactive | Fear Control | 0.56 | <0.014 | (0.07,1.05) |
| | | | No response | 0.80 | <0.001 | (0.39,1.21) |
| | | Fear control | No response | 0.24 | 0.900 | (-0.20,0.68) |

**Table 7. Showing difference in mean defensive avoidance score by efficacy threat interaction (Quadrants) of patients with diabetes in JUMC, Ethiopia April 12-May 25 2020 (n = 343).**

| ANOVA test statistics | Post hoc Method | Reference groups | Comparison group | Mean difference | P-value | 95%CI |
|---|---|---|---|---|---|---|
| F = 18.261 | Bonferroni | Fear Control | Responsive | 3.09 | <0.001 | (1.55,4.63) |
| df = 3 | | | Proactive | 2.30 | <0.001 | (0.79,3.82) |
| P-value <0.001 | | | No response | 0.76 | 0.85 | (-0.61,2.14) |
| | | No response | Responsive | 2.32 | <0.001 | (1.02,3.63) |
| | | | Proactive | 1.54 | 0.008 | (0.23,2.80) |
| | | Proactive | Responsive | 0.78 | 0.89 | (-0.65,2.22) |

dramatic/funny message were 5.2 times more likely to respond to self-care message in favor of controlling the danger compared to those who prefer fear-arousal message [AOR = 5.2(2.786, 9.706) (see Table 8).

## Discussion

This study assessed **response to self-care practice messages among patients with diabetes** in terms of the cognitive appraisal of the threat and efficacy in averting diabetes complications using the EPPM model.

This study showed that the prevalence of controlling the danger of diabetes mellitus was 49.6%. More than one-third of the respondents belong to no response group and above one-fifth of the respondents are controlling their fear of diabetes complication. Control response based on discriminatory value best predicts diabetes self-care practice. Educational status and age of the respondents have positive effect in perceived threat and perceived efficacy. More-over, responsive and proactive respondents had better diabetes self-care practice as compared to no response and fear control respondents. Different factors like educational status, informa-tion sources, and preferred message appeal, and knowledge of diabetes mellitus were predic-tors of controlling the danger of diabetes.

In this study prevalence of controlling the danger of diabetes was 49.6%, there is no finding from other studies, which supports or contradicts this finding.

More than one-third of the respondents belong to no response group: also according to fear appeal literatures [21, 24] these respondents belong to No Response i.e. People not considering the diabetes complication to be real or relevant to them; often not even aware of the diabetes

**Table 8. Predictors of Response to self-care message among patients with diabetes visiting Jimma University medical center Ethiopia, April 12- May 25, 2020.**

| Variables | Categories | COR (95%CI) | AOR (95%CI) | P-value |
|---|---|---|---|---|
| Level of Education | Cannot read and write | 1 | 1 | |
| | 1–8 | 1.24 (0.707,2.186) | 0.94 (0.466,1.901) | 0.866 |
| | 9–12 | 3.50 (1.880,6.533) | 2.74 (1.284,5.878) | 0.009* |
| | College and above | 7.55 (3.554,16.068) | 4.84 (2.016,11.612) | < 0.001* |
| preferred message appeal | Fear arousal | 1 | 1 | |
| | Dramatic | 5.49 (3.325,9.096) | 5.2 (2.786,9.706) | < 0.001* |
| Knowledge of diabetes mellitus | | 1.04 (1.026,1.051) | 1.2 (1.055,1.255) | 0.002* |
| Source of information | | 1.79 (1.475,2.179) | 1.76 (1.411,2.203) | < 0.001* |

Hosmer and Lemeshow goodness-of-fit test was chi square of 13.968 with P-value of 0.083

*Indicates significant independent predictors (p-value <0.05for characterization of perceptions toward diabetes mellitus and self-care practice among diabetes mellitus patients after adjusting all the study variables, AOR = adjusted odds ratio, COR = crude odds ratio CI = confidence interval.

complications. This shows a theory-based risk communication gap to bring about desired self-care practices in this population. Moreover, above one-fifth of the respondents are controlling their fear of diabetes complication: this are people controlling their fear by defensively avoiding to think about diabetes complication, or by reacting against it according to fear appeal literatures [21, 24]. This hampers the goal of risk communication which is moving individuals to danger control responses therefore special health risk communication needs to be developed to break through this defensive mechanism [20].

Responsive respondents had better diabetes self-care practice as compared to no response and fear control respondents. This pattern of means is consistent with the EPPM and with studies done in different parts of the world using EPPM in different contexts [24–26]. According to the EPPM, high-threatening messages coupled with high-efficacy recommendations are usually an effective means for reducing the threat (diabetes complication), and moving individuals toward protection motivation (self-care practice).

Proactive respondents had better diabetes self-care practice as compared to no response and fear control respondents. This is consistent with fear appeal literature [21] that proactive individuals are expected to demonstrate a lower level of danger control, which reinforced the EPPM's major suggestions of efficacy i.e. Perceptions of efficacy must be higher than perceptions of threat for fear appeals to be accepted by their viewer [24, 27].

In this study for a given level of perceived efficacy, variation in perceived threat did not result in a difference of self-care practice among respondents which is evidenced by the absence of difference in self-care practice between responsive and proactive respondents despite variation in threat level between the two groups and which is also evidenced by the absence of self-care practice difference between fear control and no response respondents despite this respondents had variation in threat with the same level of efficacy, furthermore proactive respondents had a better self-care practice compared with fear control respondents despite having a lower level of threat than the former respondents, this implies that in this respondents efficacy is a major determining factor which persuades individuals toward self-care practice which is supported by EPPM in which efficacy determines the nature of a response in this case diabetes self-care practice [24]. Therefore, in this population manipulation of efficacy to the highest level while maintaining the level of threat is the best that fits the existing audience's message processing to bring about desired diabetes self-care Practice.

This study reveals that fear control respondents were defensively avoidant of thinking about diabetes complication than responsive and proactive respondents which is consistent with studies done in different parts of the world with different contexts and EPPM prediction that the stronger the threat, the stronger the fear control response and the weaker the efficacy the greater the fear control response. This indicating that if fear appeals are to be used they should be accompanied by high efficacy intervention [25, 28].

According to this study, control responses based on discriminatory value best predicts diabetic self-care practice as compared control responses based on quadrants, this might be due to the use of median cut points to dichotomize threat and efficacy to high-low categories to form four quadrants resulting in misclassification of individuals close to but on opposite sides of the cut point as very different rather than very similar [29, 30].

The study revealed that respondents who completed college and university had higher odds to respond to self-care messages in favor of controlling the danger of diabetes compared to those who cannot read and write. This might be individuals with a higher educational level have better access to health-related information and can easily acquire the information they need by reading guidelines and implement professional recommendations into practice [31].

In this study increment in a score of knowledge of diabetes mellitus increases odds of controlling the danger of diabetes. This is because Self-care behaviors are the final outcome of

cognitive processes people employ during knowledge acquisition. Moreover, patients with diabetes are only willing to perform self-care behaviors when they acquire the necessary knowledge about prevention methods [31, 32].

In this study, fear arousal message had a negative effect in controlling the danger of diabetes as compared to dramatic/funny appeal, which is supported with a study conducted in Ethiopia in other contexts and with the assumption of EPPM model, which states; fear is a central variable that motivates individuals via developing defensive motivation of threat. Moreover, a message should use the appropriate appeal to persuade the receiver [17, 24]. Therefore precaution needs to be taken in communicating fear during diabetes self-care practice [33].

This study revealed that increment in the score of an information source increases the odds of controlling the danger of diabetes. This is supported by the study conducted on the repetition of the message which states that message repetition offers an audience more opportunities to scrutinize arguments and engage in systematic processing (the comprehensive analysis of a message which requires both cognitive ability and capacity) which leads to attitude changes [34].

## Strength and limitation of the study

### Strength of the study

✓ This is the first study to assess responses of diabetes patient to Self-care practice applying extended parallel process model

### Limitation of the study

To the best of the investigator's knowledge there were no similar published studies (with the same behavior) in Ethiopia, so findings were not well discussed in the related literature. Additionally, since quadrants were classified based on the median, this results in misclassification of groups, so precaution needs to be taken when interpreting and utilizing study findings. Moreover, since the data collection method was self-report rather than direct observation of the patient's self-care practice this may result in courtesy bias. However, efforts were made to minimize the bias by recruiting data collectors from other department and telling the participants about the anonymity of the data and the study have Limited generalizability because of single center study.

### Generalizability

The study will be generalized to All diabetes patients that are 18 years and above and attending follow up at diabetes mellitus clinics in JUMC.

## Supporting information

**S1 File. Questionnaire.**
(DOCX)

**S1 Data. DM eppm.**
(SAV)

## Acknowledgments

We express our heartfelt thanks to all individuals who participated in the study: respondents, data collectors, and Jimma university.

**Informed consent**

The study was ethically approved by the institutional review board (IRB) of Jimma University. The ethical clearance letter reference number is IRB 00013/20. Verbal informed consent was sought from every respondent.

## Author Contributions

**Conceptualization:** Mohammed Jemal Abawari, Yohannes Kebede.

**Data curation:** Mohammed Jemal Abawari.

**Formal analysis:** Mohammed Jemal Abawari.

**Investigation:** Zewdie Birhanu.

**Methodology:** Mohammed Jemal Abawari, Demuma Amdisa, Yohannes Kebede.

**Supervision:** Demuma Amdisa, Zewdie Birhanu.

**Validation:** Mohammed Jemal Abawari, Demuma Amdisa, Yohannes Kebede.

**Writing – original draft:** Mohammed Jemal Abawari.

**Writing – review & editing:** Demuma Amdisa, Zewdie Birhanu, Yohannes Kebede.

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
