## [Decision Letter · Decision Letter 0]

10 Aug 2021

PONE-D-21-17765

Characterization of Perceptions Toward Diabetes Mellitus and Self-Care Practice Among

Diabetes Mellitus Patients Visiting Jimma University Medical Center: Application of

Extended Parallel Process Model: Facility Based Cross Sectional Study

PLOS ONE

Dear Dr. Abawari,

Thank you for submitting your manuscript to PLOS ONE. After careful consideration, we feel that it has merit but does not fully meet PLOS ONE’s publication criteria as it currently stands. Therefore, we invite you to submit a revised version of the manuscript that addresses the points raised during the review process.

We look forward to receiving your revised manuscript.

Kind regards,

Paolo Magni

Academic Editor

PLOS ONE

Journal Requirements:

2.We note that you have stated that you will provide repository information for your data at acceptance. Should your manuscript be accepted for publication, we will hold it until you provide the relevant accession numbers or DOIs necessary to access your data. If you wish to make changes to your Data Availability statement, please describe these changes in your cover letter and we will update your Data Availability statement to reflect the information you provide

4. Please update your submission to use the PLOS LaTeX template. The template and more information on our requirements for LaTeX submissions can be found at http://journals.plos.org/plosone/s/latex

Reviewers' comments:

Reviewer's Responses to Questions

**Comments to the Author**

1. Is the manuscript technically sound, and do the data support the conclusions?

Reviewer #1: Yes

Reviewer #2: Partly

2. Has the statistical analysis been performed appropriately and rigorously? 

Reviewer #1: I Don't Know

Reviewer #2: I Don't Know

3. Have the authors made all data underlying the findings in their manuscript fully available?

Reviewer #1: Yes

Reviewer #2: No

4. Is the manuscript presented in an intelligible fashion and written in standard English?

Reviewer #1: Yes

Reviewer #2: No

5. Review Comments to the Author

Reviewer #1: Thank you for this excellent work!

I have only some comments and I hope you can do changes or explain:

1. I prefer to use patients with diabetes instead of diabetic. It means that we are setting the identity/name on people without asking them if they agree.

2. I can not find the word danger in description of the variables. I suggest not to use word danger.

3. Please send me the PDF you have in the end of the paper, I could not open it.

4. First titel in referenslist is spelled wrong: America Diabets.

Good Luck!

Reviewer #2: Jemal et al report a survey on the perceptions on how patients access, process, and respond to diabetic self-care messages in Ethiopia.

Although an interesting study, there are several major concerns with objectives, terminologies, methods, interpretation, and presentation of findings.

Major concerns:

The study objectives are not clearly stated and some are misleading. For example, the authors list: “To describe perceived threat of diabetes patient” and “�To describe perceived efficacy of diabetes patient”.

It is not clear how the survey is achieving the above when the survey was designed to assess the response to self-care message?

The authors must keep the objectives focused and achievable.

The wordings used are also somewhat incomprehensible and inconsistent. What is meant by “control the danger of diabetes? This is used in the abstract and elsewhere.

Please rephrase as appropriate.

In the methods, the authors need to specify the measuring method in the paper instead of only explaining the meaning of each measurement.

• How reliable is the “Extended Parallel Process Model” ?

• whether a high score is better or a low score is better. Has it been validated?

Highlight the limitation of a single center study and therefore may have limited generalizability even within Ethiopia

Other concerns:

Sample size:

The sample size was estimated based on the proportion of danger control response. But I cannot figure out that ‘danger control response’ is a main outcome for the analysis based on the method.

Results:

1) Page 13, last paragraph: The author reported R=0.487, please explain what analysis method was used to obtain the ‘R’.

2) Page 14, The results in Table 6 are confusing to me. How did the author define the subgroups (proactive, fear control, responsive, and no response) from Efficacy Threat Interaction? I cannot find it in the methods section.

3) Page 16, table 8:

a. How did the author define the binary outcome ‘characterization of perceptions toward diabetes mellitus and selfcare practice among diabetes mellitus’

b. Did the author only control for the factors reported in table 8 or only retained significant predictors in the model? This needs to be explained in the analysis method section. If the author only controlled for the factors in table 8 in the model, please explain why other factors were not adjusted for. Moreover, the factors in table 8 were not defined in the methods.

Please attach the survey questionnaire.

6. PLOS authors have the option to publish the peer review history of their article (what does this mean?). If published, this will include your full peer review and any attached files.

Reviewer #1: **Yes: **Marina Taloyan

Reviewer #2: No

---

## [Author Response · Author response to Decision Letter 0]

15 Sep 2021

Reviewer 1 

Dear Reviewer 1 I have read your comments and I find them helpful and make corrections based on your comments. 

Reviewer 2 

Dear Reviewer 2 I have read your comments and I find them helpful and make corrections based on your comments.

---

## [Decision Letter · Decision Letter 1]

21 Oct 2021

PONE-D-21-17765R1Response to Self-Care Messages Among Patients with Diabetes Mellitus Visiting Jimma University Medical Center: Application of Extended Parallel Process ModelPLOS ONE

Dear Dr. Abawari,

Thank you for submitting your manuscript to PLOS ONE. After careful consideration, we feel that it has merit but does not fully meet PLOS ONE’s publication criteria as it currently stands. Therefore, we invite you to submit a revised version of the manuscript that addresses the points raised during the review process.

 In particular, please include itemized response to reviewer's comments shared previously. It needs to indicate exactly where the comment was addressed (page, line number).

We look forward to receiving your revised manuscript.

Kind regards,

Paolo Magni

Academic Editor

PLOS ONE

Additional Editor Comments:

Please include itemized response to reviewer's comments shared previously. It needs to indicate exactly where the comment was addressed (page, line number)

Reviewers' comments:

Reviewer's Responses to Questions

**Comments to the Author**

1. If the authors have adequately addressed your comments raised in a previous round of review and you feel that this manuscript is now acceptable for publication, you may indicate that here to bypass the “Comments to the Author” section, enter your conflict of interest statement in the “Confidential to Editor” section, and submit your "Accept" recommendation.

Reviewer #2: (No Response)

2. Is the manuscript technically sound, and do the data support the conclusions?

Reviewer #2: Partly

3. Has the statistical analysis been performed appropriately and rigorously? 

Reviewer #2: N/A

4. Have the authors made all data underlying the findings in their manuscript fully available?

Reviewer #2: Yes

5. Is the manuscript presented in an intelligible fashion and written in standard English?

Reviewer #2: Yes

6. Review Comments to the Author

Reviewer #2: Please include itemized response to reviewer's comments shared previously. It needs to indicate exactly where the comment was addressed (page, line number)

7. PLOS authors have the option to publish the peer review history of their article (what does this mean?). If published, this will include your full peer review and any attached files.

Reviewer #2: No

---

## [Author Response · Author response to Decision Letter 1]

8 Dec 2021

Dear Plose one Academic editor 

1 A rebuttal letter that responds to each point raised by the academic editor and reviewer(s) is included we uploaded this letter as a separate file labeled 'Response to Reviewers'.

2 A marked-up copy of the manuscript that highlights changes made to the original version is uploaded as a separate file labeled 'Revised Manuscript with Track Changes'.

3 An unmarked version of the revised paper without tracked changes is uploaded this as a separate file labeled 'Manuscript'.

---

## [Decision Letter · Decision Letter 2]

13 Dec 2021

Response to Self-Care Practice Messages Among Patients with Diabetes Mellitus Visiting Jimma University Medical Center: Application of Extended Parallel Process Model: Facility Based Cross Sectional Study

PONE-D-21-17765R2

Dear Dr. Abawari,

We’re pleased to inform you that your manuscript has been judged scientifically suitable for publication and will be formally accepted for publication once it meets all outstanding technical requirements.

Kind regards,

Paolo Magni

Academic Editor

PLOS ONE

Reviewers' comments:

Reviewer's Responses to Questions

**Comments to the Author**

1. If the authors have adequately addressed your comments raised in a previous round of review and you feel that this manuscript is now acceptable for publication, you may indicate that here to bypass the “Comments to the Author” section, enter your conflict of interest statement in the “Confidential to Editor” section, and submit your "Accept" recommendation.

Reviewer #2: All comments have been addressed

2. Is the manuscript technically sound, and do the data support the conclusions?

Reviewer #2: Yes

3. Has the statistical analysis been performed appropriately and rigorously? 

Reviewer #2: N/A

4. Have the authors made all data underlying the findings in their manuscript fully available?

Reviewer #2: Yes

5. Is the manuscript presented in an intelligible fashion and written in standard English?

Reviewer #2: Yes

6. Review Comments to the Author

Reviewer #2: Jemal et al report a survey on the perceptions on how patients access, process, and respond to diabetic self-care messages in Ethiopia.

The revisions are satisfactory. I have no further comments. OK to accept.

7. PLOS authors have the option to publish the peer review history of their article (what does this mean?). If published, this will include your full peer review and any attached files.

Reviewer #2: No

---

## [Editor Report · Acceptance letter]

20 Dec 2021

PONE-D-21-17765R2 

Response to Self-Care Practice Messages Among Patients with Diabetes Mellitus Visiting Jimma University Medical Center Facility Based Cross Sectional Design Application of Extended Parallel Process Model 

Dear Dr. Abawari:

I'm pleased to inform you that your manuscript has been deemed suitable for publication in PLOS ONE. Congratulations! Your manuscript is now with our production department. 

Kind regards, 

on behalf of

Prof. Paolo Magni 

Academic Editor

PLOS ONE